# Consensus on Prioritisation of Actions for Reducing the Environmental Impact of a Large Tertiary Hospital: Application of the Nominal Group Technique

**DOI:** 10.3390/ijerph20053978

**Published:** 2023-02-23

**Authors:** Jessica F. Davies, Forbes McGain, Jillian J. Francis

**Affiliations:** 1Anaesthetics Department, Austin Health, Heidelberg, VIC 3084, Australia; 2Department of Critical Care, School of Medicine, University of Melbourne, Parkville, VIC 3010, Australia; 3Anaesthetic and Intensive Care Departments, Western Health, St Albans, VIC 3021, Australia; 4School of Public Health, University of Sydney, Camperdown, NSW 2006, Australia; 5School of Health Sciences, University of Melbourne, Melbourne, VIC 3010, Australia; 6Department of Health Services Research, Peter MacCallum Cancer Centre, 305 Grattan St., Melbourne, VIC 3000, Australia; 7Centre for Implementation Research, Ottawa Hospital Research Institute, General Campus, 501 Smyth Road, Ottawa, ON K1H 8L6, Canada

**Keywords:** consensus, environmental sustainability, healthcare, climate change

## Abstract

Hospitals are the largest greenhouse gas producers within the Australian healthcare sector due to the large amounts of energy, resource utilization, equipment and pharmaceuticals required to deliver care. In order to reduce healthcare emissions, healthcare services must take multiple actions to address the broad range of emissions produced when delivering patient care. The goal of this study was to seek consensus on the priority actions needed to reduce the environmental impact of a tertiary Australian hospital. A nominal group technique was utilized within a multidisciplinary, executive-led environmental sustainability committee to find consensus on the 62 proposed actions to reduce the environmental impact of a tertiary Australian hospital. Thirteen participants joined an online workshop during which an educational presentation was delivered, 62 potential actions were privately ranked according to two domains of ‘amenability to change’ and ‘scale of climate impact’ and a moderated group discussion ensued. The group achieved verbal consensus on 16 actions that span staff education, procurement, pharmaceuticals, waste, transport and advocacy on all-electric capital works upgrades. In addition, the individual ratings of potential actions according to each domain were ranked and shared with the group. Despite a large number of actions and varied perspectives within the group, the nominal group technique can be used to focus a hospital leadership group on priority actions to improve environmental sustainability.

## 1. Introduction

Climate change is one the greatest threats posed to human health [1]. Hospitals consume large amounts of energy, consume finite resources, use sizeable amounts of equipment and pharmaceuticals, and produce substantial amounts of waste [1]. Worldwide, the global healthcare industry accounts for approximately 5% of global greenhouse gas (GHG) emissions, and is growing year on year [1]. The international standard for adaption to climate change (ISO-14090) provides a non-linear plan for any organisation to adapt and mitigate climate change across the phases of pre-planning, assessing impacts, adaption planning and implementation (monitoring, reporting, evaluating, communicating) [2]. Eleven principles of climate change adaption are described by this standard, some of which include flexibility of the organisation to adapt and respond to new information or conditions, mainstreaming and embedding of climate change mitigation by integration into the organisation’s processes via policy, plans, procedures and implementation, and systems thinking, which ensures an understanding of the systemic issues and relationships within and beyond the organisation that influence the effectiveness of implementation efforts [2]. Although international standards for greenhouse gas emissions exist, published healthcare emissions reporting stems largely from national level emissions in high-resourced countries that are derived from economic assessments and from a small but increasing number of product-level emissions data (i.e., for specific products utilised or services delivered in healthcare settings) [3,4,5,6].

In Australia, 7% of national GHG emissions are attributed to the healthcare sector [3]. The UK’s national health service has committed to an ambitious carbon reduction target, but Australian hospitals lack coordinated national leadership, although sustainable healthcare units now exist in several Australian states (NSW, WA) [4,7]. The Victorian Government in Australia has committed to reduce the carbon footprint of healthcare [8]. Hospitals are the largest contributor of greenhouse gas (GHG) emissions within the Australian health industry and these arise from direct emissions (scope 1), indirect emissions (scope 2) and all other indirect emissions (scope 3) [9]. In the Australian healthcare industry around 60% of national emissions arise from the indirect (scope 3) emissions from 15 categories, some of which include products and services within a hospital supply chain, transport and waste [3,9]. Therefore, the actions that hospitals should take to reduce GHG emissions are broad ranging across all emission scopes but challenges exist in measuring and reporting these [10]. These actions include addressing GHG emissions from energy and buildings, transport of staff, patients and resources, emissions embedded within supply chains, waste production, eliminating low-value care, as well as leadership and engagement actions within the healthcare community itself [11]. It is obvious that, unlike in other economic sectors (such as transport), healthcare services cannot simply be replaced by a lower emissions or no-emissions alternative. Instead, the complex hospital system must be transformed by examining both clinical and non-clinical sources of waste and greenhouse gas emissions, and by preventing hospital admissions in the first place.

Nominal group technique (NGT) is a structured group consensus process which has been utilised across a variety of healthcare settings to develop consensus in clinical care, to identify research priorities, and to identify research-to-practice gaps since its original development in the 1970s [12,13]. NGT has since been applied in various settings within healthcare and research in order to reach agreement amongst stakeholders. The NGT as a method of consensus involves a structured process where ideas are generated, a moderated group discussion that ensures opinions and differing views are accommodated and a ranking or voting process [14,15,16]. Ranking within the NGT session can generate quantitative data that can be rapidly analysed, yet the discussion generated within the NGT may form the basis of further qualitative analysis [15]. As such, the NGT method combines quantitative (ranking) and qualitative (ideas generation, discussion) data collection. A strength of NGT is that the moderated discussion makes the strength of agreement and disagreement readily apparent within the group [15]. Participants are not required to prepare for the workshop but bring expertise, knowledge and experience, which in a diverse stakeholder group allows for rapid identification of feasible priorities [13]. The workshop results can also be made immediately available to the group, avoiding delays in working towards the actions required [13]. Compared to other consensus methods, it is time-effective, cost-effective, requires few resources, allows priorities to be set in only a few hours and may avoid the problems of group dynamics associated with other group methods, such as brainstorming, Delphi and focus groups [12,13,15]. Examples of NGT methods cross a range of clinical settings such as identifying priority treatment outcomes [17], developing general practice [18], identifying patient outcomes [19], identifying cancer research priorities [20], as well as guideline development as endorsed by the World Health Organisation [21]. NGT has been used to define priorities in other politicised healthcare, topics such as abortion [22], and to define the acceptability of environmental actions amongst ecology and environment stakeholders, although this was in settings outside of healthcare [23]. 

Austin Health is a large 822 bed tertiary referral health service in Victoria, Australia. In early 2022, a group of climate-concerned clinicians wrote an open letter of support to the Austin Health executive for more leadership and action on mitigating the climate impacts from healthcare. The letter was signed by over 400 staff and students of Austin Health crossing clinical and non-clinical roles. It specified the need for an executive level sustainability committee, which was agreed to in June 2022. A committee was formed through discussion between the clinician leaders and the executive group. The goal of this study was to utilise NGT in seeking consensus on the priority actions for the executive environmental steering committee at a large tertiary hospital in Melbourne, Australia, and to improve the environmental impact of healthcare delivery at Austin Health, Victoria, Australia. This activity aligns with the ISO-14090 ‘adaption planning phase’ of climate change mitigation and action at Austin Health [2]. 

## 2. Materials and Methods

### 2.1. Context

All members of the hospital’s executive environmental steering committee were invited to participate in a two-hour online workshop in September 2022, which was to be the first official meeting of the group. Prior to this, the group had opportunistically been invited to meet with an external consultancy firm, who gave a background presentation on healthcare environmental sustainability and led an interactive discussion regarding the development of a strategic direction on environmental sustainability for the hospital. Following this meeting, the draft strategic outcomes, initiatives and key performance indicators (KPIs) were shared with the group for feedback by the external consultancy firm. Although not planned, this draft strategic outcome created an important context for the workshop. 

### 2.2. Materials

Prior to the meeting, the group moderator (JD) developed a list of potential actions that may be relevant for the health service (action list) that covered all GHG emission scopes. As ISO-14090 does not provide organisation-specific guidance, but instead recommends actions be assessed as to the suitability for the organisation’s needs and capabilities, the action list was developed from four resources (see Appendix A) [2]. First were two documents produced by Doctors for the Environment Sustainable Healthcare, one titled *Net Zero Emissions: Responsibilities, pathways and opportunities* and a second that was in production prior to the meeting titled *Actions for Sustainability in your Hospital* [24]. This in-production document contained 70 actions for an executive committee and 43 actions for a clinical sustainability committee. Secondly came the Global Green and Healthy Hospitals agenda, which proposes a framework of 86 actions across 10 goals [25], and thirdly a co-located health service’s strategy titled *Caring for people and planet* provides a strategic map for the hospital to ‘achieve as close to net zero as possible’, with 137 actions for hospitals and subacute hospital sites [26]. Further, the Victorian Health Department’s sustainability strategic plan was consulted for strategy and data, and contained 21 actions [11]. The strategic actions from these four resources were combined by one researcher (JD) into one list of 71 potential actions. Although standard reporting of GHG emissions is by scope of emissions (scope 1, 2 and 3), the actions frequently crossed more than one scope, so instead this was organised across the interconnected headings of energy and buildings, transport, procurement and pharmacy, waste and leadership and engagement. The action list collation was carried out by removing duplicates and combining similar actions. 

Prior to the workshop the list of 71 collated actions was shared with a core group of environmental knowledge experts and clinical leaders at Austin Health by one researcher (JD) to ensure the action list was comprehensive and valid. This reviewing group provided additional actions or suggested changes to wording that were incorporated into the document following this consultation. Similar actions were further combined by one researcher (JD) and the resultant list contained 60 distinct actions, displayed across the three draft strategic initiatives. Workshop materials such as the collated list containing 60 actions, a run sheet, outline of the meeting program and additional resources from which the actions were sourced, were shared with the group participants prior to the meeting. These materials are presented in Appendix A.

A 30-minute background presentation was prepared by one researcher (JD) which covered carbon foot-printing in healthcare, relative impacts of components of healthcare such as energy and buildings, transport, procurement, pharmacy, food and waste, and actions for engagement and leadership. Local or Australian data were presented where possible. The presentation was pitched to be both educational for the group members who were new to environmental sustainability, yet comprehensive enough to cover all major topics represented in the action list. The presentation was structured around the draft strategic outcome, initiatives and KPIs shared with the group prior to the meeting. The slide deck prepared for the presentation is available from the corresponding author on request. 

### 2.3. Procedure

Broadly, the NGT process was conducted in seven steps during a two-hour workshop: (1) presenting background information including (2) local performance data where possible, (3) consolidating the action list presented with further potential actions, or alterations to the actions, (4) private voting on the action list according to amenability to change and scale of climate impact. Following this, (5) all participants were invited to discuss their highest-ranked actions from the list, and verbal consensus was sought through moderated group discussion (6 & 7), which corresponded to the steps described in Table 1. These steps of the NGT workshop were adapted from Rankin et al.’s (2016) adapted NGT for research-to-practice gaps [13]. Further adaptions were made following personal correspondence between the author (N. Rankin) and the principle investigator (Table 1) [13]. The adaptations included hosting online (Microsoft Teams^TM^, Microsoft Corporation, Redmond, WA, USA), screensharing a spreadsheet to prioritise instead of using paper notes, using an online platform, and omitting step 8 ‘investment exercise’, as this was felt less relevant to the topic and may not be easily delivered in an online format. 

Due to the large number of potential actions (60), further adaptions to the protocol were made to ensure timely and smooth workshop delivery. Rather than using a spreadsheet to share with the group, an online form (Google Forms, Google, Mountain View, CA, USA) was used to present the ranking form to the participants during the meeting, as this allowed the priority actions to be randomly presented and for the most rapid review of the participants’ responses. The ranking system was simplified from a numerical ranking (0–7) to ‘high’, ‘medium’ and ‘low’ [23,27]. This was because of concern regarding the time it would take to numerically rate a long list of actions across two domains. 

### 2.4. Data Analysis

For the ranking of actions, each participant’s rating for each action was assigned a numerical score (high = 3, medium = 2, low = 1) and summed separately for each factor (A; amenability to change, B; scale of climate impact). The two scores were then multiplied (A × B), and the combined score was then ranked from highest to lowest, following the precedents of Langerholc et al., and because adding the ratings together did not delineate each action sufficiently [28]. Each action was also ranked according to the rating for amenability to change and by scale of climate impact, for comparison. One out of 13 participants did not participate in voting at all and another did not complete all of the ratings so this response was excluded so as not to bias the rating. Sensitivity analysis was performed to ensure the rating process accurately reflected the verbal consensus actions. This was done by comparing the separate rankings for amenability and climate impact to the combined ranking and consensus list and comparing the impact on final scores of multiplying or adding the rankings to the consensus list. Univariate logistic regression and multivariate logistic regression were performed using R statistical software (v4.1.2, R Core Team, 2021) to determine the relationship between the individual rating scores for the variables ‘amenability to change’ and ‘climate impact’ and the odds that the action would be selected in the final agreed list of priority actions. A multivariate logistic regression was also performed to assess the relationship between each action’s combined scores of ‘amenability to change’ and ‘climate impact’ and the odds ratio of the action being selected in the final agreed list of priority actions. 

Following the meeting, the recording was transcribed and edited for accuracy. The key priorities suggested by the participants were collated into a single document and shared with the group. Field notes were recorded by the moderator during and immediately after the meeting. 

## 3. Results

### 3.1. Participant Sample

The group had varying expertise in environmental sustainability and hierarchy within the health service and represented clinical and non-clinical areas of hospital activity. The 13 members of the executive environmental steering committee included executive leadership—one of whom who had oversight of hospital infrastructure and engineering, clinician environmental leaders (consultants and one junior doctor representative), nursing leaders, a public health advocate and indigenous leader, an allied health member, an infection prevention and control consultant, a procurement and purchasing representative and the hospital sustainability officer. All members were invited, and all participated in the workshop (100% participation rate). 

### 3.2. Meeting Conduct

The meeting commenced with introductions where all participants explained their role and area of expertise in relation to environmental sustainability. Next, comprehensive background information was presented to the group by the moderator (JD). Following the background presentation there were no questions from the group members, so the priority actions were presented to the group for discussion. Following a group discussion, two further actions were added to the actions list along with some adjustment of wording, after group agreement. Next, a link to an online form was shared in the meeting and individuals completed a private ranking of each action according to the two domains of ‘amenability to change’ and the ‘scale of climate impact’. Once all private voting responses had been received, each participant was invited to share their top priorities from the list of actions they had just rated with the group. Participants were randomly called upon by the moderator (JD) and feedback was directly elicited from the relevant participant regarding feasibility or practicality of a particular action. All participants shared their top actions, although some participants presented more actions than others (up to 6 vs. 1 action). During this discussion, the moderator documented the key actions discussed. The group discussed each participant’s top actions, asked questions and sought further information from other members of the group when required. At the end of this session, after all participants had shared their actions and comments, the moderator confirmed the top priorities with the group, and clarified whether there was verbal consensus on the key actions. 

As noted in the moderator field notes, all group members actively participated for the entirety of the workshop and all participants had an opportunity to speak. Field notes indicate that the meeting discussions were respectful and perceived to be a genuine sharing of views amongst the group. In light of the range of ideas presented and discussed (although not all agreed upon), the phenomenon where participants display ‘consensus-seeking tendencies’—so called groupthink–was not observed by the moderator [29]. There was a robust discussion and differing opinions offered to most suggestions raised by participants. Although not formally elicited, feedback from the participants was positive regarding the conduct and content of the workshop session. 

### 3.3. Consensus Results

The group agreed to prioritise 16/62 actions, as presented in Figure 1. Because participants spoke broadly about the key actions and because several actions were similar, some verbally agreed priorities included more than one action. These were compiled by topic. When ranking the potential actions, the group utilised two moderator-created domains for ranking, but ultimately reached consensus on key actions that were not always the highest ranked on either paradigm, nor overall, through discussion and agreement (Table 2). Appendix A contains the complete list of ranked actions. 

The most commonly agreed upon priority related to hospital-wide education on environmental sustainability. This was raised by one participant, who said that one of the biggest issues to overcome was how education and training of internal and external staff could be achieved, such that the staff members understood that nearly every decision they take has an environmental repercussion. After agreeance, an additional potential action was added that reflected the need for mandatory staff training. Many participants’ top priorities related to other ‘easy wins’. ‘Easy win’ actions were those actions that the participants perceived to be highly feasible, and that may have added benefits, such as building momentum and community engagement in environmental sustainability activities. In terms of scale of climate impact, there was little discussion about the relative impacts of each action and there was no dispute about whether a proposed action would have a positive environmental impact. The group discussions valued the importance of ‘bringing others along with them’, as demonstrated by the high value placed on education, communication and engagement actions in discussions. Embedding actions into the hospital staff activities and more broadly into future strategy was referenced by many participants. The group discussion often touched on the importance of practical operationalising of the actions themselves, that is, frameworks, strategies and governance structures that would facilitate individuals to make the best environmental decisions in their day-to-day practice. Much of the discussion around prioritising procurement-related actions included the need for baselines and reporting standards as well as the importance of informed, scientific, financial and environmental decision making. Discussions concerning actions that had longer time frames, such as all-electric new capital works, required in depth knowledge to be able to action or were advocated for, but not necessarily directly under the hospital management control, as the final approval did not always lay with the hospital itself (i.e., Department of Health).

Throughout the discussions, the main reason for disagreement was feasibility. The most commonly disagreed upon actions related to infrastructure, buildings and upgrades. This was largely through further facilities, management or financial information coming to light through group discussion. For example, although capital upgrades that were all-electric were agreed upon by the group as very important for improving the carbon footprint of the health service, decision making around this was usually undertaken by those external to the hospital management and was therefore deemed to be more suited to advocacy rather than to a committee level action. Further, it was suggested that actions relating to clinical practice be separated from those more directly related to hospital management decisions, such as electric vehicle procurement and energy use; however, separating clinical from non-clinical actions within the group was difficult, as many overlap and reflect the complex systems that hospital staff work within. 

Eight of the top 20 ranked priorities were also ranked in the top 20 for both amenability to change and for impact on climate. Of the top 20 ranked priorities, 14 were also in the top 20 priorities when ranked only by amenability to change, which was the same number that were also in the top 20 when ranked only by climate impact. There was greater concordance between the dual-ranked proposed actions and the rating for amenability to change rather than impact on climate. Overall, the actions were more commonly ranked as high rather than low. Of the verbally agreed consensus actions (Figure 1), seven of the top priorities mentioned by participants related to education, six related to procurement and/or reusable equipment, four related to pharmaceuticals and four to waste. Three top priorities related to electric vehicles and one priority was for a wider approach to transport emissions in general. Three participants brought up choosing wisely and/or low-value care and clinical behaviour changes. The most commonly mentioned top priorities relating to education and procurement were also in the top 27 rankings, out of 62 actions. Univariate and multivariate odds ratios (ORs) for both the cumulative scores of each factor and the combined product of both factor scores did not show significant correlation with being selected as an agreed priority action (Table 3), suggesting that there were other factors at play (such as group discussion) that were a greater influence on whether an action was selected by the group as an agreed priority. 

After the study was completed, a further meeting of the group endorsed the recommendations with no amendments, and this was taken as a confirmation of the validity of the consensus process.

## 4. Discussion

In this study, the nominal group technique (NGT) was employed in a multidisciplinary group of hospital leaders to achieve consensus on actions to improve the environmental sustainability within a tertiary hospital in Australia, in line with the adaption planning phase of the international standard of climate change adaption. The top priorities were to educate the workforce on climate change and health, improve procurement processes to prioritise reusable equipment and reduce packaging waste, improve waste services including addressing food and clinical waste, reduce pharmaceutical impacts, introduce a package of travel improvements to and within the hospital, and to advocate for all-electric new building projects. There was little correlation between the numerical score assigned by each participant in the ranking phase of the NGT with whether the action was likely to be selected by the group for prioritisation. This reflects the mixed-methods technique of NGT as it is likely that other factors such as group discussion and participant interaction led to the decisions made by the group, rather than a numerical rating alone. This project demonstrated that NGT can be performed in a two-hour online workshop and consensus on environmental sustainability actions can be achieved despite a large and diverse scope of clinical and non-clinical actions. 

One strength of this study is that the group consisted of clinician experts in environmental sustainability in healthcare coupled with non-expert hospital leaders, who were able to reach agreement on actions deemed to be either amenable to change, or have a significant climate impact, or both. That the numerical scores alone did not show correlation with whether an action would be selected as a priority action demonstrates the strength of the group discussion. It also endorses the need for participants with institutional knowledge, as well as ‘green’ clinician leaders, to collaboratively discuss their relevant insights in a group setting. Failure to reach consensus may otherwise delay action on reducing healthcare’s impact on climate change. Likewise, including executive leadership did not impair the nominal group technique for consensus, and likely enhanced the group’s ability to find feasible actions that the group could deliver beyond the factors of amenability to change and impact on climate change. 

Further, while the international standard 14090—Adaption to Climate Change provides a generalised and non-linear model for any organisation to adapt and to mitigate the impact of climate change, and although it recognises the complex systems thinking required to undertake environmental sustainability activities, it does not provide specific steps relevant to the healthcare industry [2]. Additionally, ISO-14090 emphasises measurement and reporting of GHG emissions, but the largest source of emissions in healthcare are scope 3 (indirect) emissions and assessment of these are underdeveloped, lacking in product-level data and lacking in normalisation factors for healthcare services of differing sizes and scopes of practice [2,10]. Other sustainability resources provide comprehensive lists of actions that hospitals could take to reduce their impact on climate change, yet long lists may be overwhelming and distract from the actual business of delivering institutional change. As more healthcare services take actions to limit their environmental impact, this study demonstrates that even a large number of actions can be successfully prioritised. 

Interprofessional education, feedback and local opinion leaders are the most effective means of improving uptake of evidence-based practice changes [30,31,32]. Undertaking a consensus finding workshop can provide an opportunity to deliver interprofessional education on climate change to hospital leaders, could foster the creation of feedback systems, and can be driven by institutional ‘green champions’, thus meeting the increasing calls for clinician leadership on healthcare transformation [7,33]. One example of successful interprofessional education was highlighted by the high ranking of the action relating to reducing nitrous oxide leaks. Typically, nitrous oxide wastage is the niche domain of select clinicians whose practice may utilise the greenhouse gas (anaesthetists, emergency physicians, obstetricians in obstetric hospitals), but this action was unexpectedly ranked highly at 10/44. This is attributed to the educational content of the background presentation. Although this study does not purport to measure the impact of the workshop beyond the achievement of consensus, these are potential benefits that may generate enthusiasm and momentum for those health services looking to improve their environmental credentials. 

One limitation of this study is that it only represents one large tertiary public hospital in Australia and experiences in the private health sector may be different. Additionally, the increasing public support for greater action on climate change, coupled with an increasing awareness of the interplay between climate change and health amongst health professionals, may in part be responsible for the success of this workshop, particularly as the qualitative components of the NGT are more likely to be the driving factors in prioritising an action [34,35]. Further, the context of this study changed rapidly in the lead up to the workshop. This study benefited from the environmentally focused hospital strategic plan that was made clear to all participants prior to the workshop. Sharing of this strategic plan may have led the participants to perceive the hospital management to be more invested and committed to delivering real environmental improvements, and this may have increased the participants’ perceptions that an action is feasible and amenable to change. Hospitals without board-level commitment to act on climate change may find a similar consensus workshop more challenging to deliver. In addition, although participants were asked to rank the actions according to the action’s perceived climate impact, a worldwide survey of healthcare workers found that healthcare workers were keen to improve their engagement with environmental sustainability in healthcare, but lacked knowledge to do so [35]. With a different group composition, the agreed priority actions may have differed. ISO-14090 recognises that ‘organisations are complex and that an assessment of which potential climate actions are best suited to the organisation’s needs and capability’ can be performed using contextually suitable decision making methods [2]. A challenge for all healthcare sustainability actions is that most actions lack data to quantify any climate benefits for nearly all of these actions, and where it does exist (such as product-level life-cycle assessment) it requires careful interpretation, because results can be dependent on location and therefore local energy sources. Additionally, not all actions can be easily delineated by greenhouse gas emission scopes, or might not even relate directly to quantifiable emissions reduction (such as staff education), but were still deemed as top priorities of the group because of recognition of a wider institutional need [9]. Despite lacking data to support all the actions, it is unlikely that a high impact action was excluded in the list, because none were rejected by the action list validation nor the participant group, and all additional suggestions were included in the action list. Finally, a number of changes were made to the adapted version of the nominal group technique described by N. Rankin et al. [13] for this unique setting, and these adaptions need validation in similar contexts.

## 5. Conclusions

A coordinated approach to delivering high-value care to patients that has the least environmental detriment can be embedded throughout a health service. To begin this complex process, stakeholders must agree on the priority actions to target. However, hospitals, like the planet, are complex environments where the implications of one action may have other unknown consequences, which may in turn have unintended positive or negative implications for other interconnected factors. For example, one environmentally driven action may influence workforce morale, workforce movements throughout the hospital, have unknown financial costs or benefits and changes to water usage or carbon emissions, and these may not be known or even measurable. Despite this, we conclude, based on the findings from this study, that it is possible to achieve consensus, even among staff with widely different perspectives, on actions to reduce the carbon footprint of healthcare. The nominal group technique can be used to focus a hospital leadership group on priority actions to improve the health of patients and the planet.

## Figures and Tables

**Figure 1 ijerph-20-03978-f001:**
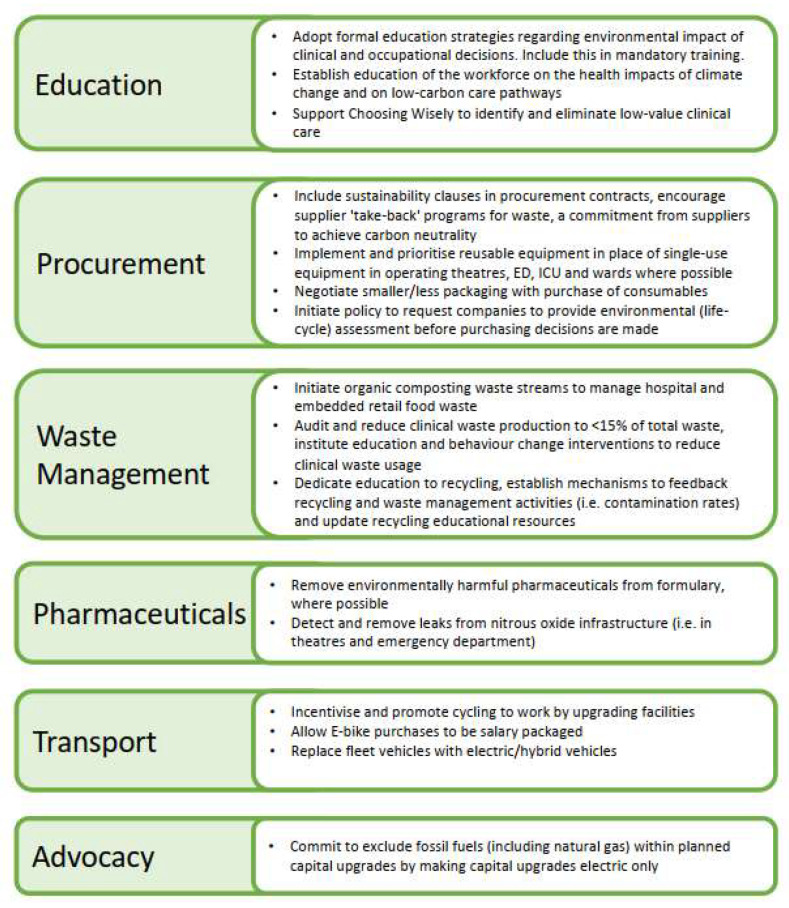
Verbal consensus on priority actions for Austin Health executive sustainability steering committee.

**Table 1 ijerph-20-03978-t001:** Nominal group technique steps described by Rankin et al. (2016) and modifications made to nominal group technique steps in finding group consensus for environmental sustainability actions.

Steps	Rankin et al. (2016) [13]	Environmental Sustainability Group Consensus	Time Frame
1	Describe research to practice gaps	Describe background information on healthcare environmental impacts	30 min
2	Present local data/information about the gaps	Present local data on baseline environmental performance, where available	
3	Elicit feedback and record additional gaps identified by participants	Elicit feedback and record additional actions identified by participants	10 min
4	Individuals vote privately to prioritise gaps, using moderator-created criteria	Individuals vote privately to prioritise gaps according to two moderator-created criteria: - Amenability to change - Scale of climate impact	10 min
5	Each participant selects the two most important gaps from the prioritised list	Each participant is invited to share their top priorities from the list of 60	30 min
6	Focus group participants discuss ratings and moderator uses matrix tool as a tally sheet	Whole group discussion discussing the top priorities, feasibility, past history of the institution in relation to the action, where available. Moderator keeps tally of priorities brought up by individuals	
7	Whole group discussion and consensus	Verbal consensus on top priority actions for the group	15 min
8	‘Investment exercise’ of how each participant would spend $100,000	Collation of remaining actions according to ratings	15 min (if time)

**Table 2 ijerph-20-03978-t002:** The top 20 ranked potential actions a tertiary Australian hospital could take to improve the environmental sustainability of the health service from 11 participant ratings of high (=3), medium (=2) or low (=1) according to amenability to change and the action’s climate impact, with summed ratings (A and B) and the total score (product of A and B).

Final Rank	Proposed Priority Actions	Amenability to Change Summed Ratings (A) [n = 11, Possible Score Range 11–33]	Climate Impact Summed Ratings (B) [n = 11, Possible Score Range 11–33]	Ranking Total Score (A × B)
		Mean Score (1–3) [High = 3 Medium = 2 Low = 1]	Mean Score (1–3) [High = 3 Medium = 2 Low = 1]	
**1**	Engage hospital communications team to increase messaging around sustainability, promote local initiatives and broadcast any wins/successes in order to build momentum	32 3.0	28 2.3	896
**2**	Expand telehealth supports between Austin teams and referring health services to decrease the need for transfers and promote facilitation of care at regional/rural facilities	29 2.5	29 2.7	841
**3**	Include sustainability clauses in procurement contracts, encourage supplier ‘take-back’ programs for waste, a commitment from suppliers to achieve carbon neutrality	28 2.7	29 2.5	812
**4**	Reduce patient travel with improved technology to support telemedicine and appointment bundling	27 2.7	30 2.5	810
	Establish local area green groups led by local area green champions to coordinate initiatives across areas, share resources and educate	27 2.6	30 2.4	810
**5**	Commit to exclude fossil fuels (including natural gas) within planned capital upgrades by making capital upgrades electric only	29 2.6	27 2.6	783
**6**	Establish education of the workforce on the health impacts of climate change and on low-carbon care pathways	26 2.8	30 2.1	780
	Printing: double sided printing as standard on all printer settings, mandate paper sources to be ethically and sustainably sourced, work towards models of care that are paperless	26 2.5	30 2.4	780
**7**	Lights off at night program: Reduce energy usage after hours including clinical areas with low night-time activity and offices	31 2.8	21 2.5	775
**8**	Implement and prioritise reusable equipment in place of single-use equipment in operating theatres, ED, ICU and wards where possible	27 2.5	28 2.4	756
**9**	Reduce environmental hazards by auditing use of Hydrofluorocarbons (HFCs), Sulpha hexafluorides (SF6s) and explore the possibility to reduce, prevent wastage or replacement	26 2.5	29 2.4	754
	Initiate energy power-downs of high-energy usage areas such as heating ventilation and cooling (HVAC) in theatres during times of reduced activity such as nights and weekends	29 2.3	26 2.7	754
**10**	Detect and remove leaks from nitrous oxide infrastructure (i.e., in theatres and emergency department)	25 2.7	30 2.3	750
	Advocate for fossil-fuel free (all electric), zero emission and high-efficiency approaches to all new capital works	30 2.5	25 2.7	750
**11**	Consider strategic sustainability research projects that will lead to financial and environmental savings for the hospital	27 2.5	27 2.5	729
	Coordinate a triple-bottom-line decision-making policy for introducing new single-use equipment, through liaison with infection control, procurement and stakeholders	27 2.3	27 2.4	729
**12**	Reduce energy use by installing motion-controlled lighting, occupancy sensors, LED lighting	26 2.5	28 2.4	728
	Reduce single use disposable plastics in embedded retail	26 2.5	28 2.6	728
**13**	Adopt a strategy to reduce/replace fossil fuel including natural gas use within existing buildings, and where they cannot be immediately replaced optimise their efficiency	29 2.5	25 2.6	725
**14**	Mandate that all bidding contractors have net zero approach to capital works	30 2.2	24 2.5	720
	Adopt formal education strategies regarding environmental impact of clinical and occupational decisions. Include this in mandatory training.	24 2.6	30 2.6	720
**15**	Initiate sustainable pharmaceutical procurement for example, by preferencing companies with a net zero emissions target	27 2.2	25 2.1	675
**16**	Dedicate education to recycling, establish mechanisms to feedback recycling and waste management activities (i.e., contamination rates) and update recycling educational resources	24 2.5	28 2.3	672
	Negotiate smaller/less packaging with purchase of consumables	28 2.5	24 2.5	672
**17**	Embed sustainability into research plans and funding such as requiring environmental assessments in new research approvals and fostering research and innovation through dedicated grant funding	25 2.6	29 2.1	667
**18**	Audit and reduce clinical waste production to <15% of total waste, institute education and behaviour change interventions to reduce clinical waste usage	26 2.4	25 2.3	650
	Initiate policy to request companies to provide environmental (lifecycle) assessment before purchasing decisions are made	26 2.2	25 2.5	650
	Implement reusable PPE across all sites	26 2.3	25 2.5	650
	Source hospital and embedded retail foods from local and sustainable food sources	26 2.0	25 2.2	650
**19**	Utilise electric vehicles for inter-facility shuttle transport	24 2.6	27 2.5	648
**20**	Commit to no piped nitrous oxide in future building or capital works upgrades	25 2.5	25 2.3	625
	Initiate campaign to reduce contamination of co-mingled recycling, and provide single-stream recycling in appropriate areas (i.e., soft plastics in ward areas)	25 2.3	25 2.2	625

**Table 3 ijerph-20-03978-t003:** Univariate and multivariate odds ratio (OR) of the likelihood that the numerical rating score for an action correlated with that action being selected as a priority action.

Score	Univariate OR (95% CI)	Univariate *p*	Multivariate OR (95% CI)	Multivariate *p*
Amenability to change (A)	1.03 (0.88, 1.22)	0.734	0.98 (0.81, 1.17)	0.799
Climate impact (B)	1.23 (1.02, 1.56)	0.05	1.24 (1.02, 1.59)	0.052
Total (A × B)	1 (1, 1.01)	0.187	N/A	N/A

## Data Availability

Data is provided in Appendix A.

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
