# Peer review of "Consensus on Prioritisation of Actions for Reducing the Environmental Impact of a Large Tertiary Hospital: Application of the Nominal Group Technique"

_ijerph, 2023, doi:10.3390/ijerph20053978_

Round 1

Reviewer 1 Report

I have read with interest this interesting manuscript by Davies et al., where the authors report the methodology and results of a nominal group technique approach to seek consensus on the priority actions to reduce the environmental impact of a tertiary Australian hospital.

The article is well-written, and the methodology and outcomes are adequately presented. 

In order to improve the manuscript, some minor comments should be addressed before publishing:

Introduction section:

-. Line 57: The number of hospital beds should be informed.

-. The last two paragraphs in this section are redundant and a bit confusing. There seem to be two goals, while the main goal is described in the first paragraph (lines 82-84: The goal of this study was to utilise NGT in seeking consensus on the priority actions for the executive environmental steering committee at a large tertiary hospital in Melbourne, Australia, with a view to reducing the environmental impact of the organisation.) That is in line with the goal depicted in the Abstract section. In order to better clarify for the reader, I suggest removing lines 85-88, and simply adding something like "and improve the environmental impact of healthcare delivery at Austin Health, Victoria, Australia."

Supplementary file:

-. The table in the Word document should have a heading, indicating what it depicts. In addition, there is a comment in the table that should be removed before publishing the supplementary material.

Reviewer 2 Report

Dear authors, in order to improve the quality of your work, you need to:
1. Describe in considerable detail the method you used, with more examples of applications in the same or similar fields,
2. Correct the technical part of the text - blank pages are not allowed,
3. Review system standards that can be used to define priority activities in this area, such as ISO 14000 series standards, specifically 14060 and 14090,
4. Create a detailed, quantified discussion. Although relatively simple, the NGT method allows for various correlation analyses, single, multiple, etc.
5. On a related previous note, add few advanced visualization methods to present research results, don't rely exclusively on tables.
Best Regards

Round 2

Reviewer 2 Report

Dear authors,

thank you for valuable improvement of your paperwork.

Kind regards